# Research on Fire-Detection Algorithm for Airplane Cargo Compartment Based on Typical Characteristic Parameters

**DOI:** 10.3390/s23218797

**Published:** 2023-10-28

**Authors:** Haibin Wang, Hongjuan Ge, Zhihui Zhang, Zonghao Bu

**Affiliations:** 1Civil Aviation College, Nanjing University of Aeronautics and Astronautics, Nanjing 211106, China; allenge@nuaa.edu.cn; 2Civil Aviation Safety Engineering College, Civil Aviation Flight University of China, Guanghan 618307, China; zhangzhihui@cafuc.edu.cn (Z.Z.); fy00014167@cafuc.edu.cn (Z.B.)

**Keywords:** PSO-LSTM, GA-RF, BP-AdaBoost

## Abstract

To clarify the reasons for inaccurate fire detection in aircraft cargo holds, this article depicts research from the perspective of a single type of sensor detection. In terms of fire smoke, we select dual-wavelength photoelectric smoke sensors for fire-data collection and a genetic algorithm to optimize the classification and detection of random forest fires. From the perspective of fire CO concentration, we use PSO-LSTM to train a CO concentration compensation model to reduce sensor measurement errors. Research is then conducted from the perspective of various types of sensor detection, using the improved BP-AdaBoost algorithm to train a fire-detection model and achieve the high-precision identification of complex environments and fire situations.

## 1. Introduction

According to the U.S. Federal Aviation Administration (FAA) Technology Center, the aircraft fire alarm rate is only 0.5%, and the false-alarm rate is as high as 99.5% [1,2]. A statistical analysis of China’s civil aviation safety information shows that, between 2014 and 2022, there were 11 alarms for fire and smoke accidents and 73 alarms for general types of fire and smoke incidents, but no real-fire alarms occurred [3,4]. The Airworthiness Standard for Transport Aircraft (CCAR-25-R4) [5] of Part 25 of the Civil Aviation Regulations of China stipulates that a fire-detection system must detect a fire in the cargo compartment within one minute and warn the crew. With the increasing cabin sizes of civil aircraft, the cargo hold height can reach more than four meters, and it is difficult for smoke to reach the top detector position in a short time. To realize a timely alarm, detector sensitivity can be improved; however, this indirectly increases the occurrence of false alarms. At the same time, because the crew cannot conduct a visual check during the flight to confirm a false alarm, to reduce the false-alarm rate, the cargo hold fire-detection system must be improved [6].

In single-parameter fire-detection technology, in view of the interference of various aerosols on fire detection in aircraft cargo holds, Zheng Rong analyzed the asymmetric ratio variation of different types of aerosols, developed a dual-wavelength smoke detector principle validator, and differentiated studies of interfering aerosols [7]. Chen et al. used a four-frame-difference method to extract smoke images from active infrared cameras. By calculating the image frame sequence to extract the texture features of smoke and set the threshold of smoke fire detection, they could meet the airworthiness requirements of transport aircraft and effectively improve the speed of fire-smoke detection [8]. Liu Feng et al. designed a smoke concentration detection system based on Mie theory, using a photoelectric detector to detect laser signals scattered by smoke particles and convert them into electrical signals. They found a quantitative relationship between the transmitted and scattered light intensity of the laser signal scattered by the smoke. This study demonstrated a high sensitivity for detecting smoke concentration [9]. Smith proposed a signal control system to detect fire, using a microprocessor to identify the gas, temperature, smoke concentration, and UV, and experimentally proved its reliability [10]. Meyer et al. conducted experiments on the combustion smoke characteristics of five typical aircraft materials on the International Space Station. The response time of the sensor to different smoke types was measured by controlling the heating temperature, air flow rate, and heating time, and they studied and compared the detection effects of the smoke detector on various types of smoke in different pressure environments [11]. Ukleja et al. studied the characteristics of smoke changes both in and out of a corridor-type confined space during a fire. They found that the smoke concentration on the side of the exhaust pipe decreased, and the smoke concentration on the side of the confined space increased after a fire; however, the air flow pattern in the whole space was reversed after the air flowed out from one side of the space [12]. Vasiliev et al. proposed an early fire-detection method for large airflow spaces, such as aircraft cargo holds [13].

In multi-parameter fire-detection technology, Sucuoglu et al. used smoke, flame, and temperature sensors to detect fires, designed a data-fusion fire system, and trained the data fusion using the least squares method, thereby improving fire-detection accuracy to 92% [13]. Mahhub et al. designed an embedded fire-detection system that can collect and store humidity, temperature, CO_2_, and smoke fire characteristic parameters and transmit data in real-time to a cloud server through HTTP protocol for monitoring anytime and anywhere [14]. Nnaemeka et al. designed an indoor fire-detection and monitoring system based on wireless sensor networks, with a battery-monitoring scheme that can greatly enhance the system’s endurance [15]. Brian et al. integrated infrared imaging, distributed chemical sensors, and smoke detectors through digital signal processors to determine fires, thereby greatly improving the accuracy of fire detection [16]. Robert et al. trained a multi-sensor-fusion fire-detection algorithm by combining convolutional neural networks with fuzzy logic and proposed a fire-detection and alarm system suitable for various scenarios, with an accuracy rate of over 90% [17].

The following elements are the limitations of the above studies:(1)A lack of collection and analysis of characteristic parameters of typical combustibles under variable pressure. The fire characteristic parameters of combustibles in smoldering or open fires are obviously different, and the cargo hold of an aircraft is different from that of a general indoor environment. The characteristic parameters of smoldering or open fire of combustibles under different atmospheric pressure conditions vary greatly, which is an important cause of false alarms. Hence, their study is helpful to analyze their fire behavior characteristics and improve the accuracy and sensitivity of fire detectors.(2)Multi-sensor-fusion fire-detection systems need further research. To reduce the false-alarm rate, many researchers choose one or more characteristics, such as that of smoke, gas, and temperature, as the basis of judging fire; however, traditional fire sensors cannot meet the requirements of accurate detection in the complex environment of an aircraft cargo hold.

We use genetic algorithms to optimize the classification and detection of random forest smoke data. We use PSO-LSTM to train a CO concentration compensation model to reduce the measurement error of the sensor, integrate multi-source data, and train fire-detection models using the BP-AdaBoost algorithm to improve testing accuracy.

## 2. Detector and Experimental Design

### 2.1. Dual-Wavelength Detection

Dual-wavelength smoke detectors reflect fire-smoke characteristics by the ratio of the received optical power to the transmitted optical power (PTR) [17]. Particle concentrations can be characterized by the PTR, which differs by wavelength. We use blue light at λ1 = 470 nm and red light at λ2 = 870 nm as the incident light source. Red light is used to measure the volume concentration of smoke particles and blue light is used to measure their surface area concentration.

The surface area and volume concentrations of smoke particles are proportional to the PTR and have a similar trend of change. We introduce the Sotter average particle size here, as follows: [17]
(1)$$D_{32}=6\times \frac{T_{S}}{T_{V}}\times \frac{P_{V}}{P_{S}}$$
where $P_{V}$ is the red PTR; $P_{S}$ is the blue PTR; $T_{V}$ is the conversion coefficient between the blue light scattering intensity and the volume concentration of smoke, which reflects the light scattering intensity of the volume concentration of particles per unit volume; and $T_{S}$ is the conversion coefficient between the scattering intensity of red light and the surface area concentration of particles, which reflects the scattering intensity of the surface area concentration of particles per unit volume.

In the actual measurement, because the scattered light signal is usually converted into an electrical signal for sampling, the conversion coefficients $T_{S}$ and $T_{V}$ are affected by circuit parameters such as LED-light-emitting efficiency, detector-light-receiving efficiency, and the gain coefficient of the amplifier circuit. According to the response of the dual-wavelength sensor used in this research, the ratio conversion coefficient $\frac{T_{S}}{T_{V}}$ is selected to be 100.

The ADPD188BI dual-wavelength detector is an integrated optical module for smoke, which contains two optical detectors. A 470-nm blue light source and 870-nm red light source drive the LED to emit light and measure the corresponding return light signal. The device can read measurements either directly from the output register or through a first-in-first-out (FIFO) buffer. The device was wired as shown in Figure 1.

The red and blue PTRs of typical combustibles smoldering in a closed space are detected, and the Sauter mean diameter is calculated. To simulate the environment of an airplane cargo compartment, a square experimental cabin of 2 m × 2 m × 2 m was used as the combustible test environment.

According to the “Technical requirements and test methods for point smoke detectors” (GB4715-2005 [7]) and the FAA’s minimum performance standard for Airborne Halon as a substitute for fire-extinguishing agents, corrugated paper, cotton rope, and beech wood (100 mm × 100 mm × 4 mm) were used as the fire source, where the grain direction was the chord direction, the length of the cotton rope was 240 cm and the diameter was 7 mm, and the corrugated paper sheet was 200 mm × 200 mm × 1 mm.

A heating furnace was used to ignite the combustibles in the center of the bottom of the shelter. For quantitative analysis, 20 g of combustible material was placed in the center of the furnace for uniform heating and ignition in each experiment. Each experiment lasted 20 min, with the initial furnace heating the combustibles at full power from room temperature for 10 min at a constant rate and closing for 10 min. The dual-wavelength smoke sensor was located 1.5 m above the heating furnace. Its dual-wavelength emission module circuit board was parallel to the ground. The light source faced down toward the combustible and was used to collect three kinds of solid combustibles, from non-fire to smoldering. Then, the PTR values of the red and blue light and the change trend of the average particle size of set were measured. Cameras were set up to monitor combustion. The experiments were conducted three times to avoid experimental error or special cases.

### 2.2. CO Detection

We investigated the following four kinds of CO sensors with different detection principles: semiconductor, electrochemical, catalytic combustion, and non-dispersive infrared. Table 1 lists the relevant sensor parameters, advantages, and disadvantages. The semiconductor CO sensor detects gas resistance at a high temperature, can work stably only when kept at about 400 °C for a long time, and consumes much energy. Therefore, it was excluded. The catalytic combustion CO sensor uses a strong catalyst to make combustible gas ignite on its surface. It has a high working temperature and large error in an anoxic environment, and, hence, was also excluded. The non-dispersive infrared sensor uses an infrared ray emitted by gas molecule vibration, according to the gas absorption of a specific wavelength of infrared, to detect CO, in which the infrared transmittance of the gas concentration is measured. This sensor has high precision but is more suitable for detecting the concentration of pure gas. The large amount of smoke produced by a fire can disturb and even block the light path, which leads to sensor failure. Therefore, this sensor is not suitable in confined spaces. The electrochemical CO sensor is based on the principle of a fuel cell. It generates a redox current proportional to the CO concentration. It has the characteristics of zero power consumption, a short response time, and a long life, therefore, we selected this as the detector in our study.

The CO detection experiment platform was consistent with dual-wavelength smoke detection. A heating furnace was used to ignite the combustible materials in the center of the shelter bottom, and an electrochemical carbon monoxide concentration sensor was arranged in the center of the shelter top, perpendicular to the heating furnace. For each experiment, 20 g of combustible material was placed in the center of the furnace for uniform heating and ignition. Each experiment lasted 20 min. The furnace heated the combustibles at full power from room temperature for 10 min at a constant rate, and then shut down for 10 min. The carbon monoxide concentration, environmental temperature, and humidity data were collected during the entire process. Each experiment was repeated three times to avoid the effects of experimental errors or special cases.

## 3. Single Sensor Fire Detection

### 3.1. Optimization of Smoke Fire Detection in Random Forest Based on Genetic Algorithm

The genetic algorithm (GA), borrows from the natural law of “survival of the fittest, survival of the fittest of the fittest”, simulating the natural evolution process to search for an optimal solution. It can represent complex structures by simple encoding and can form a population by encoding the initial parameters; implement genetic operations such as selection, crossover, and mutation to screen chromosome data; and iterate until a condition is satisfied.

Random forest (RF) is an integrated learning algorithm whose basic units are decision trees. It builds a random forest by randomly and repeatedly sampling K samples from the original training sample set N to generate new training sample sets and uses these to generate K classification trees. The classification results of the new data are determined by voting on the classification trees, which are constructed independently. Due to the random feature selection method, each node divides different features randomly and compares the error in different cases to select the best feature. Although the classification power of a single tree may not be high, by performing a statistical analysis of the classifiers for each tree, we can select the most likely classifiers, thereby improving the classification accuracy. The essence of random forest is to fuse multiple decision trees into one tree, where each tree has the same distribution, and its correlation determines its classification performance. The eigen estimation error, classification ability, and correlation degree also determine the number of selected features. A GA is used to optimize the number of decision trees and binomial trees, so as to improve the accuracy of the classification model.

The classification model is shown in Figure 2, and is built as follows:(1)The experimental data are randomly divided into training and test sets, and the two-wavelength PTR, blue PTR, and Sauter mean particle size, measured by the two-wavelength smoke detector, are used as the classification input values;(2)The number of decision trees and binomial tree variables in the RF algorithm are set as optimization parameters to initialize the population, and the population size and dimension, iteration times, and crossover and mutation probability are determined;(3)In each iteration, the best fitness, average fitness, and best chromosome of the population of each generation are obtained and recorded;(4)The number of decision tree and binomial tree variables is determined when the algorithm iterates, until its fitness meets the requirement, or it reaches the maximum number of iterations;(5)The optimal parameters are input into the RF algorithm for classification training and model building;(6)The accuracy rate, precision rate, recall rate, and F1 value are introduced to evaluate the classification effect of the model.

In this experiment, the combustibles can be divided into two cases: no fire and fire. MATLAB2019B was used as the training platform of the algorithm, and the RF algorithm was used to train a binary classification model of the experimental data. The accuracy, precision, recall, and F1-score are often selected to evaluate classification algorithms. Because this research includes fire and non-fire samples as positive and negative samples, respectively, we define these terms as follows:(2)$$\mathrm{Accuracy}=\frac{TP+TN}{TP+TN+FP+FN}$$
(3)$$Precision=\frac{TP}{TP+FP}$$
(4)$$Recall=\frac{TP}{TP+FN}$$
(5)$$F1\text{-}score=2\times \frac{Precision\times Recall}{Precision+Recall}$$
where TP is the number of correctly classified positive samples, FP is the number of negative samples classified as positive, FN is the number of positive samples classified as negative, TN is the number of correctly classified negative samples, *Accuracy* is the proportion of correctly classified samples to all of the samples, *Precision* is represented in the results of the predicted positive samples and the true percentage of positive samples, *Recall* is the percentage predicted to be positive in the actual positive samples, and *F*1-*score* is based on a harmonic average of precision and recall, ranging from 0 to 1, where results closer to 1 indicate that the overall classification effect is better.

In this study, 1400 data items were obtained by processing and filtering the experimental data. A total of 1000 of these were randomly selected as a training set, and the remaining 400 items were used as a test set. The final classification results are shown in Table 2.

Comparison and analysis show that the GA is more accurate than the SVM and BP neural networks, being about 9% higher than the SVM and 5% higher than the BP neural network. This analysis shows that the overall classification accuracy of this algorithm is obviously better than the other two algorithms, and the precision of this algorithm and BP is much higher than SVM’s 83.61%. Furthermore, the smoke characteristic parameters of typical solid combustibles can be distinguished clearly in the case of no fire and smoldering, and the false-alarm rate of the two algorithms is very low in the case of fire-smoke detection. The proposed algorithm and SVM perform better than BP, which shows that the false-alarm rate of fire detection can also be significantly reduced. Finally, the *F*1-*score* algorithm is also much better than the SVM and BP, showing that GA-RF is better than SVM and BP in every aspect. Through the overall analysis, it can be seen that the GA-optimized RF algorithm has a good effect on the classification of non-fire and smoldering combustibles in a closed space and shows great potential for the combination of dual-wavelength smoke detectors for fire detection in aircraft cargo compartments.

### 3.2. CO Concentration Compensation Algorithm Based on PSO-LSTM

After processing and screening, a total of 1054 groups of experimental data were obtained. After random scrambling, 5 groups of data were randomly selected from each of the three combustibles as the model training test set, and the remaining 1039 groups were used as the training set. The test set data are shown in Table 3.

Considering that the accuracy of the measurement of the electrochemical carbon monoxide detector is easily affected by changes in the ambient temperature and humidity, the measurement of the electrochemical carbon monoxide detector real-time measurements of the ambient temperature and humidity are input variables in the training model. The types of combustibles burned are classified as a class of input variables. With continuous combustion, the ambient temperature in the simulated cabin rises slowly, the ambient humidity decreases gradually, and the error between the measured and standard values increases. Therefore, the variation of ambient temperature and humidity is the key cause of the error of the electrochemical CO detector. A neural network is used to train the predictive compensation model, so as to reduce the error of the electrochemical CO detector.

Although the LSTM solves the problem of gradient vanishing and explosion in recurrent neural networks, the introduction of a gate structure increases the number of model parameters—whose setting is usually based on the experience of researchers—which has a certain subjectivity and must be adjusted manually in order to find the optimal parameters. Particle swarm optimization (PSO) is used to determine the optimal key parameters, which enable the best prediction effect. Klaus Greff et al. discussed the setting of LSTM-related hyperparameters. The experimental results show that the learning rate is the most important parameter in the LSTM, followed by the network size, and the descending gradient has little influence on the final results.

The number of hidden layer neurons and the learning rate of the LSTM model were optimized by PSO, and then the model of carbon-monoxide-fire detection and compensation was established by using the optimal parameters. The compensation model is built is as follows, as shown in Figure 3:The experimental data were randomly divided into training and test sets, where the values of the electrochemical carbon monoxide detector were used as the measurements and the CO concentration of the flue gas analyzer as the standard values. Then, the data were normalized;The number of network neurons and the learning rate of the LSTM were set as the optimization parameters, the particle swarm was initialized, and the population size, population dimension, number of iterations, learning factor, limit range of velocity, and location of individual populations were determined;The prediction error of the PSO-LSTM model was taken as the fitness value of particles and taken as the global extreme value. The individual extremum was updated repeatedly according to the change in the fitness value. If the individual extremum of the whole particle swarm was better than the global extremum, then the latter was replaced;The number of hidden neurons and the learning rate were determined when the algorithm iterated until the fitness was stable or the maximum number of iterations was reached;The optimal parameters were input into the LSTM training forecast to train the network and establish the model;The R2, mean absolute percentage error (MAPE), and root mean square error (RMSE) were used to evaluate the prediction effect of the model.

The goodness of fit, average absolute percentage error, and root mean square error were used to evaluate the reliability and accuracy of the training model. The fitting degree of the standard curve of the CO concentration was evaluated by R^2^, which is between 0 and 1, where a value closer to 1 indicates a higher fitting degree of the model. This is defined as follows:(6)$$R^{2}=1-\frac{\sum_{i}{\left({\hat{y}}_{i}-y_{i}\right)}^{2}}{\sum_{i}{\left({\bar{y}}_{i}-y_{i}\right)}^{2}}$$
where y_i_ and ${\hat{y}}_{i}$ are the respective predicted and standard values of sample *i*. MAPE reflects changes in particle fitness, in which case the lower its value, the smaller the percentage error of the predicted value relative to the standard value, and the better the error compensation effect. The RMSE represents the degree of deviation between the predicted and standard values. A smaller RMSE indicates a smaller difference between the result and the true value, and a better training effect. The MAPE and RMSE are defined as follows:
(7)$$MAPE=\frac{100\%}{n}\sum_{i=1}^{n}\left|\frac{{{\hat{y}}_{l}-y}_{i}}{y_{i}}\right|$$
(8)$$RMSE=\sqrt{\frac{1}{n}\sum_{i=1}^{n}{\left(y_{i}-{\hat{y}}_{l}\right)}^{2}}$$
where n is the number of samples.

After continuous adjustment and optimization, the initial parameters of the particle swarm were as follows: 100 maximum iterations, particle swarm size n = 20, local search ability c1 = 2, and global search ability c2 = 2. At this point, the number of hidden neurons was 19 and the learning rate was 0.1058. Figure 4 and Figure 5, respectively, show the trends of the particle swarm algorithm’s optimization of the number of hidden layer neurons and the learning rate with iterations. The error compensation results of PSO-LSTM on the test set data were compared with the compensation effects of LSTM, BP, and GA-BP, as shown in Figure 6.

From Figure 6, it can be seen that the measured values of electrochemical carbon monoxide sensors are generally higher than those of the standard values recognized by flue gas analyzers, and the volume fraction between the two continues to expand with an increasing concentration. The increase in carbon monoxide concentration indicates that the fire is constantly growing, causing an increase in the environmental temperature and a decrease in environmental humidity, which is an important reason for the increased error of electrochemical carbon monoxide sensors. By comparing the compensation effects of the four algorithms, it can be seen that the overall compensation effect of the PSO-LSTM exceeds that of the other three algorithms.

Table 4 shows the error evaluation results of four neural networks, from which it can be seen that R^2^ is close to one for all of the algorithms, indicating that their fitting degree has reached a high level, and there is no underfitting or overfitting that leads to insufficient or excessive training and model failure. The MAPE and RMSE of PSO-optimized LSTM are better than those of the unmodified LSTM, the BP neural network, and the GA-optimized BP neural network. The MAPE is about 6% less than that of the unmodified LSTM, 3% less than that of the unmodified BP neural network, and about 1% less than that of GA-BP.

It can be seen that the computation times of the BP and GA-BP are relatively similar, as are those of the LSTM and PSO-LSTM. However, the overall time consumption of the two algorithms based on LSTM is slightly greater than that of the two algorithms based on the BP neural network. Through comparison, it can be seen that the PSO-optimized LSTM algorithm has a higher solution accuracy, smaller error, stronger generalization ability, and a good compensation effect on the electrochemical carbon monoxide detector.

## 4. Multi-Parameter-Fusion Fire-Detection Algorithm

The traditional BP-AdaBoost model is based on the AdaBoost ensemble learning concept, using the BP network as a weak learner to establish a more powerful binary classification learner. The algorithm flowchart is shown in Figure 7.

We study the fire discrimination of non-fire, smoldering fire, and open fire, which can be used in three or more classifications, and has the following steps:(1)The initial distribution weight is determined by the number of samples as follows:
(9)$$D_{t}\left(i\right)=\left(w_{11},\ldots ,w_{1i},\ldots ,w_{1N}\right),w_{1i}=\frac{1}{N},i=1,2,3,4,\ldots N$$
where $D_{t}(i)$ is the distribution weight of sample i in turn t, and N is the number of samples;(2)When training the weak learner in group t, the training data are used to train the neural network and predict the output of the training data as follows:
(10)$$e_{t}=\sum_{i}D_{i}\left(i\right)=1,2,\cdots ,m\left(g\left(t\right)\neq y\right)$$
where g (t) is the predicted output and Y is the expected output;(3)There are m weak classifiers, m = 1, 2, 3, …, M. The basic classifier is obtained by learning from the training dataset of Dm with weight distribution as follows:
(11)$$G_{m}\left(x\right):x\to \left\{1,2,3\right\}$$(4)After training with a weak classifier, the classification error rate of $G_{m}\left(x\right)$ on the training data set is calculated as follows:
(12)$$e_{m}=\frac{1}{N}\sum_{i=1}^{N}R\left(G_{m}\left(x_{i}\right)\neq y_{i}\right)$$(5)The sequence weight is calculated based on the classification error rate as follows:
(13)$$a_{m}=\frac{1}{2}\ln\frac{1-e_{m}}{e_{m}}$$(6)The next training sample weight is calculated according to the sequence weight as follows:
(14)$$D_{m+1}\left(i\right)=\frac{D_{t}\left(i\right)}{B_{t}}\times exp\left[-a_{m}f_{i}G_{m}\left(x_{i}\right)\right]$$
where $f_{i}\in \left\{-1,\;1\right\}$. If the prediction is correct, $f_{i}$ = 1 and otherwise $f_{i}$ = −1. *B_t_* is a normalization factor, as follows:(15)$$B_{t}=\sum_{i=1}^{N}D_{m+1}\left(i\right)exp\left(-a_{m}f_{i}G_{m}\left(x_{i}\right)\right)$$(7)The strong learner function is calculated from the weak learner function, as follows:
(16)$$G\left(x\right)=round\left(\frac{1}{M}\sum_{t=1}^{T}a_{t}G_{m}\left(x\right)\right)$$
where $G_{m}\left(x\right)$ is a function trained by a weak learner.

We established a BP-AdaBoost fire-detection classification and recognition algorithm using MATLAB2019B. The inputs for model training were red light PTR (volume concentration), blue light PTR (surface area concentration), temperature, and CO concentration. The output was the fire status for identification and judgment, i.e., no fire, smoldering, or open fire.

The algorithm was evaluated by accuracy, precision, recall, and F1-score, as follows:(17)$$Accuracy=\sum_{i=1}^{k}\frac{{TP}_{i}+{TN}_{i}}{{TP}_{i}+{TN}_{i}+{FP}_{i}+{FN}_{i}}$$
(18)$${Precision}_{i}=\frac{{TP}_{i}}{{TP}_{i}+{FP}_{i}}$$
(19)$${Recall}_{i}=\frac{{TP}_{i}}{{TP}_{i}+{FN}_{i}}$$
(20)$${F1\text{-}score}_{i}=2*\frac{{Precision}_{i}*{Recall}_{i}}{{Precision}_{i}+{Recall}_{i}}$$
(21)$$Macro-Precision=\frac{1}{k}\sum_{i=1}^{k}{Precision}_{i}$$
(22)$$Macro-Recall=\frac{1}{k}\sum_{i=1}^{k}{Recall}_{i}$$
(23)$$Macro-F1\text{-}score=\frac{1}{k}\sum_{i=1}^{k}{F1\text{-}score}_{i}$$

The index i=1,2,3,⋯,k indicates that the classification algorithm must perform k types of classification on the samples, where k = 3 in this case, indicating no fire (i = 1), smoldering (i = 2), or open fire (i = 3).

${TP}_{i}$ is the number of samples that correctly classify the i-th sample, ${FP}_{i}$ is the number of samples misclassified as class i, ${FN}_{i}$ is the number of samples misclassified as not being in class i, ${TN}_{i}$ is the number of samples correctly classified as not being in class i, *Accuracy* is the overall proportion of correctly classified samples, ${Precision}_{i}$ is the percentage of correctly classified i-th samples, ${Recall}_{i}$ is the percentage predicted to be in class i among the actual samples in class i, ${F1\text{-}score}_{i}$ is based on the harmonic mean of precision and recall for the i-th class of samples, Macro−Precision is the arithmetic mean of the sum of all types of precision, Macro−Recall is the arithmetic mean of the sum of all types of recall, and Macro−F1-score is the arithmetic mean of the sum of all types of F1-scores.

After processing and filtering the experimental data, 7000 pieces of experimental data were obtained. After random shuffling, 6000 were selected as the training set, and the remaining 1000 were selected as the testing set. The final classification effect is shown in Table 5.

Through comparative analysis, it can be seen that, compared to a single BP neural network and the AdaBoost algorithm, the improved BP neural network based on the AdaBoost algorithm principle can be used as a weak classifier in AdaBoost; furthermore, when combined with a strong AdaBoost classifier, it can effectively detect, classify, and recognize complex combustible fire states in simulated aircraft cargo hold pressure-changing closed environments. The overall classification accuracy is over 96%, which is about 3% higher than that of the BP neural network, AdaBoost, and SVM. For the accuracy of smoldering, the improved algorithm is about 6% higher than that of the first two, and about 4% higher than that of SVM. The improved algorithm has a fire-free recall rate that is about 14% higher than that of the BP neural network, 16% higher than that of AdaBoost, and 7% higher than that of SVM. The macro precision and macro recall of the improved algorithm are higher than those of the other three algorithms, indicating good application effects in all aspects. Coupled with the multi-sensor-fusion aircraft cargo fire-detection system designed in the previous section, it can have huge potential.

The comparison in the Table 6 shows that single-parameter fire detection, which uses two-wavelength data to describe the particle size of smoke particles, can reach an average of 90.91% accuracy after applying the algorithm, and the F1-score of the algorithm can reach a maximum of 0.9142, whereas, the multi-parameter fire-detection method can reach an average of 93.92% accuracy, and the average F1-score of the algorithm can reach an average of 0.9397. In summary, when comparing single- and multi-parameter fire detection, the accuracy of fire detection using multiple parameters is higher, as is the quality of the model.

## 5. Conclusions

We proposed improvements to address the issue of high false-alarm rates caused by traditional photoelectric smoke detectors in civil aircraft cargo holds, which are susceptible to interference from factors such as smoke, dust, and water vapor. Using an experimental cabin to simulate the aircraft cargo environment, the fire-detection performance of typical fire sensors was studied, from which we drew the following conclusions:(1)For the smoke characteristic parameters of fires, we selected a dual-wavelength photoelectric smoke detector and established an experimental platform to measure and collect three parameters—red light PTR, blue light PTR, and Sotter average particle size—during the smoldering process of typical solid combustibles. A genetic algorithm was used to optimize a random forest for fire classification and recognition of collected data, achieving good recognition results, and effectively distinguishing it from interference sources. Regarding the smoke characteristic parameters of fires, we studied the changes in the CO concentration during the combustion process. By researching and selecting common CO sensors on the market, the CO concentration changes during combustible fire processes were collected on an experimental platform. Combined with the CO concentration measured by a smoke analyzer, PSO-optimized LSTM was selected to compensate for the concentration error of the electrochemical CO detector, which can effectively improve its measurement accuracy.(2)The improved BP-AdaBoost algorithm, which uses a BP neural network as a weak classifier and AdaBoost as a strong classifier, trained the fire-detection classification model on the measured data. The training results indicate that the model could effectively distinguish and classify various typical combustibles in variable-pressure environments and has great application prospects.

## Figures and Tables

**Figure 1 sensors-23-08797-f001:**
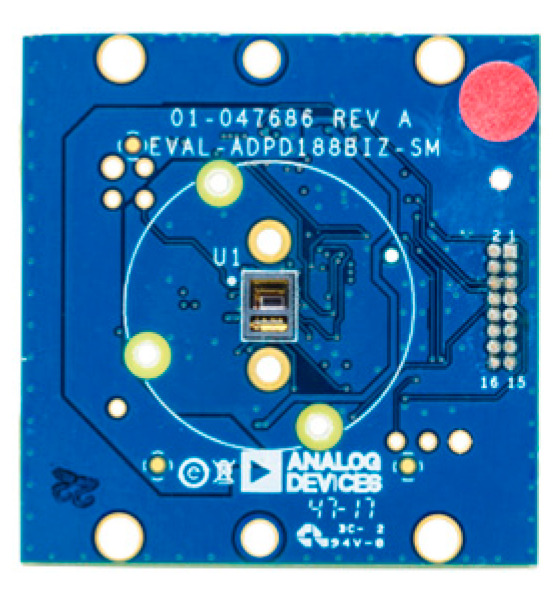
ADPD188BI dual-wavelength smoke detector.

**Figure 2 sensors-23-08797-f002:**
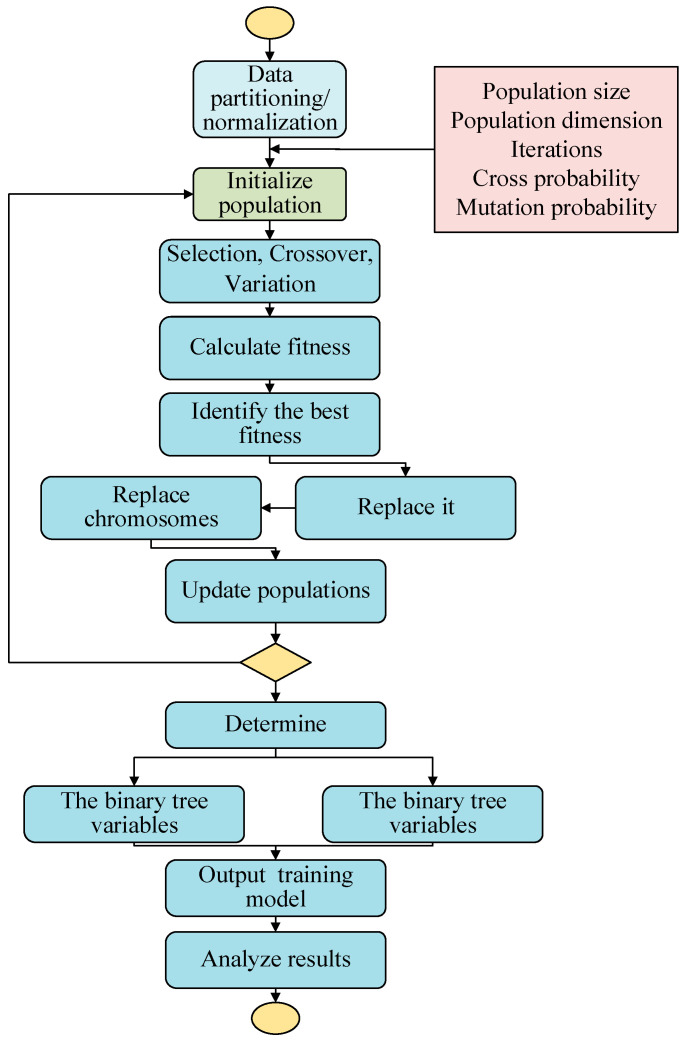
GA-RF principle flow chart.

**Figure 3 sensors-23-08797-f003:**
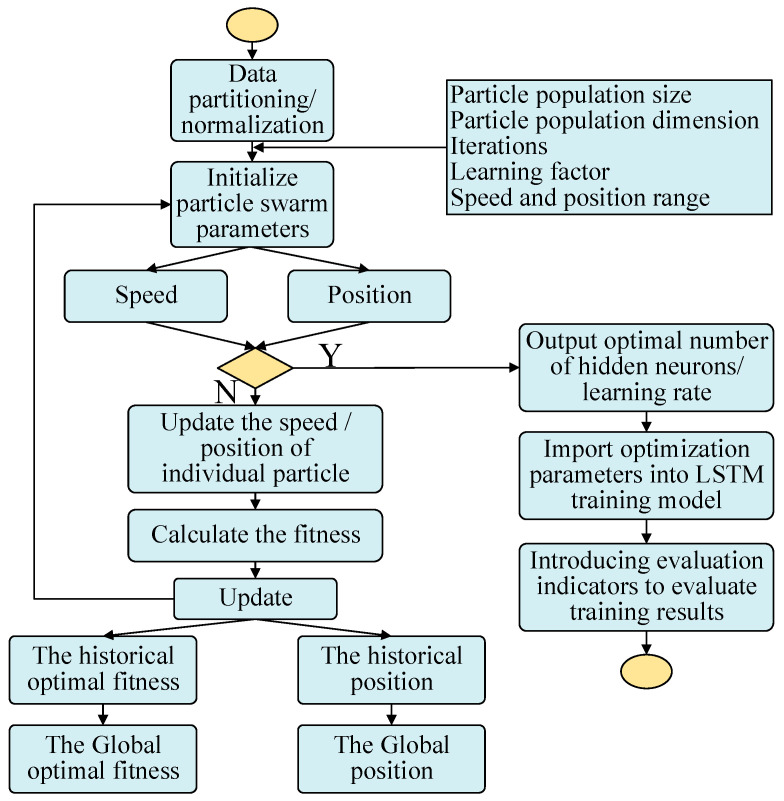
PSO-LSTM process.

**Figure 4 sensors-23-08797-f004:**
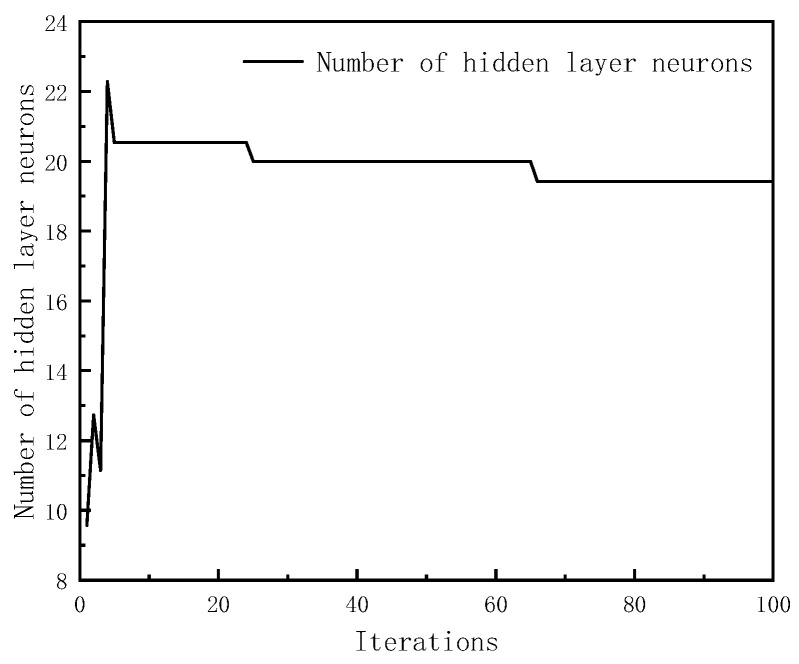
Optimization of the number of neurons in the hidden layer.

**Figure 5 sensors-23-08797-f005:**
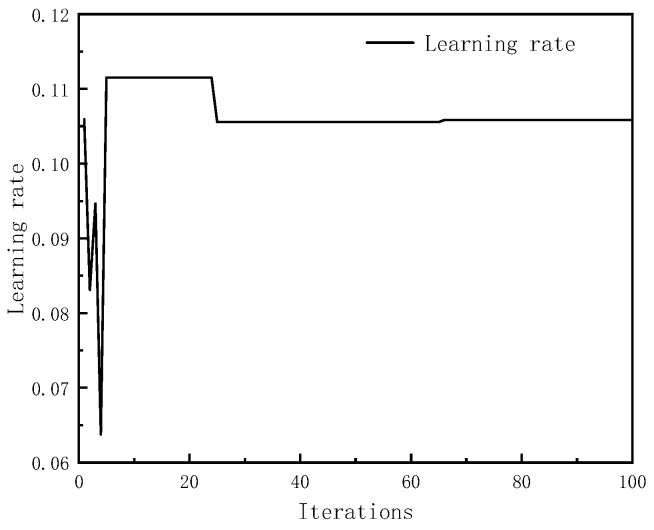
Optimization Change of Learning Rate.

**Figure 6 sensors-23-08797-f006:**
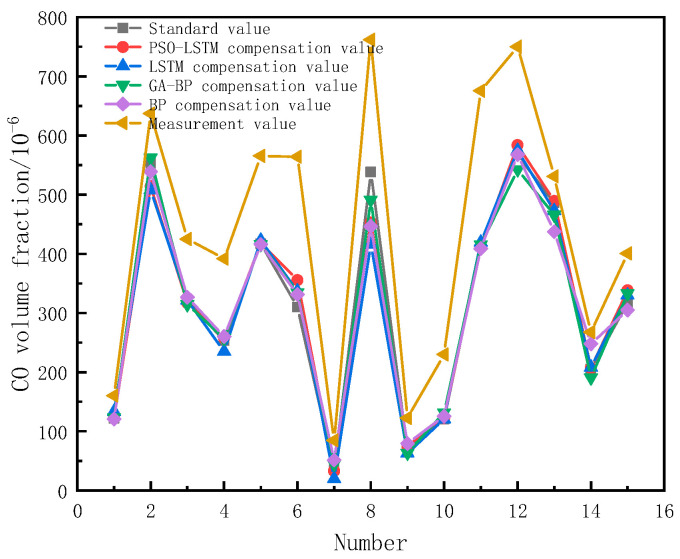
Comparison of Training Effects.

**Figure 7 sensors-23-08797-f007:**
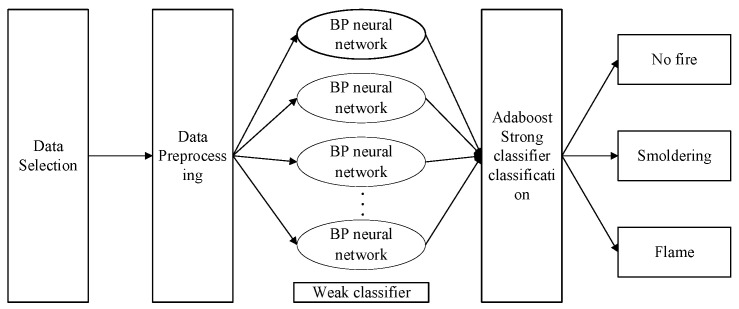
BP-AdaBoost algorithm flowchart.

**Table 1 sensors-23-08797-t001:** Comparison of CO sensor models.

Model	Working Temperature/ Relative Humidity	Range	Response Time	Advantages	Disadvantages
Semiconductor type	0 °C–+50 °C\55% ± 5%RH	10~500 PPM (75 PPM)	<60 s	High sensitivity, long lifespan, low cost, and simple driving circuit	High power consumption and susceptible to temperature, humidity, airflow, etc.
Electro-chemical	20 °C~50 °C\15%~90%RH	0~1000 PPM (1 PPM)	<30 s	High sensitivity, good stability, good linearity, fast response speed, and long service life	Not suitable for exposure to alcohol, paint, and other environments, which can affect its lifespan and accuracy
Catalytic combustion type	−40~+70 °C, ≤95%RH	0~100 (%LEL) linearity (%) ≤ 5%	<40 s	The output signal is close to linearity, with good stability, accurate measurement, and fast response	The presence of sulfides and hydrogen in the detected gas has a significant impact on the detection accuracy, and most organic vapors have toxic effects on the sensor
NDIR (Non-Dispersive Infrared)	−10~60 °C 0~95%RH	0–10,000 PPM (1 PPM)	<30 s	Wide range, high accuracy, good selectivity, high reliability, no adsorption effect, no poisoning, no dependence on oxygen, less environmental interference, and long service life	High price, high maintenance difficulty, large volume, unsuitable for portable instruments, low concentration detection accuracy needs to be improved, and it is not suitable for long-term power supply use

**Table 2 sensors-23-08797-t002:** Results of classification algorithm evaluation.

Algorithm	Accuracy	Precision	Recall Rate	F1-Score
GA-RF	95.75%	94.50%	96.67%	0.9557
SVM	86.67%	83.61%	96.00%	0.8938
BP	90.33%	93.50%	85.50%	0.8932
None	70.70%	77.00%	64.00%	0.6990

**Table 3 sensors-23-08797-t003:** Test Dataset.

NO	Measurement Value/ppm	Ambient Temperature/°C	Ambient Humidity/RH	Material Category	Standard Value/ppm
1	160.3	28.8	42.9	Beech	122
2	636.9	32.9	33.0	Beech	553
3	425.0	31.0	39.3	Beech	322
4	391.9	30.0	41.3	Beech	253
5	565.5	32.3	36.7	Beech	417
6	563.9	28.7	37.0	Paper	310
7	84.6	28.2	45.0	Paper	35
8	761.8	29.4	37.6	Paper	538
9	122.5	28.2	35.2	Paper	69
10	230.1	28.5	45.6	Paper	121
11	675.6	27.3	54.7	Cotton rope	413
12	750.0	29.5	48.8	Cotton rope	567
13	530.8	29.9	39.0	Cotton rope	486
14	267.6	26.0	55.8	Cotton rope	205
15	400.7	29.1	39.1	Cotton rope	317

**Table 4 sensors-23-08797-t004:** Neural Network Error Evaluation Results.

Algorithm	R^2^	MAPE	RMSE
PSO-LSTM	0.98530	4.4501%	7.2437
LSTM	0.96230	10.664%	35.5557
BP	0.97593	7.8364%	30.3702
GA-BP	0.99120	5.6498%	17.6504

**Table 5 sensors-23-08797-t005:** Evaluation Results of Fire-Detection Classification Model.

Evaluating Indicator	BP	AdaBoost	SVM	BP-AdaBoost	None
Accuracy	0.93255	0.93353	0.92458	0.96617	0.70321
No-fire-detection accuracy	0.97727	0.97674	0.95403	0.97566	0.68014
Smoldering-detection accuracy	0.89293	0.89827	0.91818	0.95327	0.75178
Open-flame-detection accuracy	0.95226	0.95833	0.94188	0.95804	0.65383
No-fire recall rate	0.82692	0.80769	0.90154	0.97436	0.71422
Smoldering recall rate	0.96305	0.95843	0.91536	0.95234	0.77785
Open-flame detection rate	0.94009	0.95392	0.94848	0.96700	0.65577
No-fire F1 value	0.89583	0.88421	0.90774	0.95410	0.65048
Smoldering F1 value	0.92667	0.92737	0.94664	0.95774	0.73782
Open-flame F1 value	0.95105	0.95612	0.94500	0.95249	0.69218
Macro precision	0.94416	0.94445	0.92470	0.95233	0.71592
Macro recall rate	0.91002	0.90668	0.92179	0.96454	0.68021
Macro F1 value	0.92452	0.92257	0.93313	0.95811	0.69346

**Table 6 sensors-23-08797-t006:** Evaluation of Single- and Multi-parameter fire detection.

Type	Accuracy	F1-Score
Smoke	90.91%	0.9142
Multi-parameter fire detection	93.92%	0.9397

## Data Availability

All relevant data are within the paper.

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
