# Peer review of "Research on Fire-Detection Algorithm for Airplane Cargo Compartment Based on Typical Characteristic Parameters"

_sensors, 2023, doi:10.3390/s23218797_

Round 1
Reviewer 1 Report
This paper conducts research on the key technologies of typical fire detectors. Using genetic algorithm is used to optimize random forest fire classification detection; Using PSO-LSTM to train a CO concentration compensation model to reduce sensor measurement errors; Using the improved BP AdaBoost algorithm to train the fire detection model. The method you proposed is interesting and feasible. However, the overall creativity of the algorithm is not particularly sufficient. Secondly, there are many details in the algorithm part that have not been explained clearly, which need further explanation.
1. In the first paragraph after the formula on the third page of the article, there is an error in the parameter TV.
2. It is suggested to complete the chapter serial number of the title.
3. The serial number (1) of the sixth page of the article is not aligned with other serial numbers
4. In GA-RF,do population size, Cross probability, Mutation probability and other parameters remain unchanged in the experiment? If unchanged, it is necessary to consider whether its different parameter settings will affect the experimental results. If the impact is small, it needs to be explained.
5. The GA-RF algorithm does not explain what its fitness is represented by.
6. The fourth point on the sixth page of the article refers to the satisfaction of the conditions or the number of iterations. The article does not explain what the conditions are. If there are no other conditions, does the algorithm only need to meet the requirements of the number of iterations?
7. In this paper, 12 pages, the four algorithms for the consumption of time from where to see? Need to supplement the figure or table.
8. Line 419, 420: The parameter does not explain the formula of line 418.
9. The parameter setting problem when comparing all algorithms.
10. All tables need to unify the question of whether they are centered, and the cross-page problem of tables.
minor editing
Author Response
The article has been revised based on expert feedback, please review again. Thank you very much.

Reviewer 2 Report
1. There are big problems in the layout of this paper, such as the table does not exceed the page, the icon is not centered, etc.
2. Abstract description is too redundant, and the highlights and significance of this paper need to be further expressed clearly.
3. The drawings drawn in this paper are rather sloppy, so it is suggested to redraw them to increase the readability of the paper.
4. Can more advanced models be added to this paper for comparison?
5. There are few references in this paper, and article in the past two years should be cited more frequently, such as:
(1) BCMNet: Cross-Layer Extraction Structure and Multiscale Downsampling Network With Bidirectional Transpose FPN for Fast Detection of Wildfire Smoke.
6. The sentences in this article need further polishing.
no
Author Response

(The authors gave the same response as above.)

Reviewer 3 Report
Addressing the issues of high false alarm rates by traditional smoke detector in civil air craft cargo, this paper explores 1) optimization of signals from a dual wave length photoelectric smoke detectors by using genetic algorithm for random forest fire classification and recognition of the collected data; 2) optimization of carbon monoxide detector data by using Particle swarm optimization (PSO) LSTM; 3) development of a multi parameter fusion fire detection algorithm.
The topic of the paper is very interesting, but the paper requires major revision for the paper organization, methodology, test set-up description, result presentation, and result discussion.
Here are my review comments:
1. While in the introduction of the paper, the author pointed out the issue of high false alarm rates caused by nuisance sources, the paper actually only discusses the accuracy of detection, and the paper does not consider the false alarm issues caused by nuisance sources. The introduction of the paper should be re-written for the clarification.
2. In Table 3.2, readers won’t be able to see the improvement made by those three algorithms. Please provide accuracy, precision results before applying any of those algorithms. So that one can see the benefits of using an algorithm.
3. In table 4.1, please also include evaluation results without any of those processing (done by BP, Adaboost, SVM etc.), So that one can see the benefits of the data processing.
4. Multi parameter fusion fire detection algorithm:
Please provide descriptions for the input data used in the multi parameter fusion algorithm development. I assume that both data from the dual wave length photoelectric smoke detectors and CO detector are used, but the paper does not clearly explain what was done in this section.
5. Multi parameter fusion fire detection algorithm:
Also, provide a section to discuss the performance of the multiple parameter fusion algorithm in comparison to that of the single sensor fire detection. This could be easily done using the results already presented in the paper.
6. There are sentences that are either too long, unclear, or incomplete. Please re-write them.
7. Many symbols used in all equations are not clearly explained. Please make them all clear.
For example, D32 in Eq 3.2, and there are many errors in the descriptions of the equation from 4.1 to 4.15.
8. There are typos: for example, the ‘sotter’ mean particle size in page 3
9. Also references are weird: what is ”[64]” in page 3.
1. There are sentences that are either too long, unclear, or incomplete. Please re-write them.
Author Response

(The authors gave the same response as above.)

Round 2
Reviewer 2 Report
accept
Reviewer 3 Report
Glad to find that authors addressed all the comments and made the changes as per my review.